# Glutathione contributes to efficient post-Golgi trafficking of incoming HPV16 genome

**Shuaizhi Li**[1], **Matthew P. Bronnimann**[1¤], **Spencer J. Williams**[2], **Samuel K. Campos**[1,2,3,4]*

**1** Department of Immunobiology, University of Arizona, Tucson, AZ, United States of America, **2** Department of Molecular & Cellular Biology, University of Arizona, Tucson, AZ, United States of America, **3** Cancer Biology Graduate Interdisciplinary Program, University of Arizona, Tucson, AZ, United States of America, **4** BIO5 Institute, University of Arizona, Tucson, AZ, United States of America

¤ Current address: Roche Tissue Diagnostics, Tucson, AZ
* skcampos@email.arizona.edu

## Abstract

Human papillomavirus (HPV) is the most common sexually transmitted pathogen in the United States, causing 99% of cervical cancers and 5% of all human cancers worldwide. HPV infection requires transport of the viral genome (vDNA) into the nucleus of basal keratinocytes. During this process, minor capsid protein L2 facilitates subcellular retrograde trafficking of the vDNA from endosomes to the Golgi, and accumulation at host chromosomes during mitosis for nuclear retention and localization during interphase. Here we investigated the relationship between cellular glutathione (GSH) and HPV16 infection. siRNA knockdown of GSH biosynthetic enzymes results in a partial decrease of HPV16 infection. Likewise, infection of HPV16 in GSH depleted keratinocytes is inefficient, an effect that was not seen with adenoviral vectors. Analysis of trafficking revealed no defects in cellular binding, entry, furin cleavage of L2, or retrograde trafficking of HPV16, but GSH depletion hindered post-Golgi trafficking and translocation, decreasing nuclear accumulation of vDNA. Although precise mechanisms have yet to be defined, this work suggests that GSH is required for a specific post-Golgi trafficking step in HPV16 infection.

## Introduction

Human papillomavirus (HPV) is the most common sexually transmitted infection in the United States [1]. Currently there are >400 HPV types that have been deposited and annotated in the Papillomavirus Episteme (PAVE) database [2]. Based on disease association, clinically relevant HPV can be divided into low-risk HPV, causing cutaneous and mucosal warts, and cancer-associated high-risk HPV [3,4]. High-risk HPV are associated with 99% of cervical cancers and 5% of all human cancers worldwide [5,6]. HPV16 belongs to the high-risk HPV, and HPV16 alone is responsible for >50% cervical cancers [7]. There are highly effective vaccines being used to prevent high-risk HPV infection, but the vaccine cannot protect against all types of high-risk HPV infection and the high cost prevents people from developing world to get access to the vaccine [7,8].

**Data Availability Statement:** All relevant data are within the manuscript.

**Funding:** SKC is supported by grant 1R01AI108751-01 from the National Institute for Allergy and Infectious Diseases, https://www.niaid.

nih.gov/. The funders had no role in study design, data collection and analysis, decision to publish, or preparation of the manuscript.

**Competing interests:** The authors have declared that no competing interests exist.

HPVs are small non-enveloped DNA viruses. The icosahedral capsid is built from 72 pentamers (360 molecules) of the major capsid protein L1 [9]. Within this particle, variable copies (<72 molecules but typically 20–40) of minor capsid protein L2 are complexed to the ~8kb dsDNA genome (vDNA) [10,11]. At the cellular level, HPV infection begins with virion attachment to keratinocytes via heparin sulfate proteoglycans (HSPGs) followed by furin- and kalleikrein- dependent cleavage of the capsid [12,13], likely causing conformational changes and transfer to secondary entry receptor complex(es). The nature of the entry complex(es) is still debatable but likely include growth factor receptor tyrosine kinases, integrins, tetraspanins, and annexin A2 [14,15]. Asynchronous cellular uptake of virion occurs by macropinocytosis-like mechanisms after this potentially prolonged residence on the cell surface [16,17].

Internalized virions initially traffic through the endosomal compartment where low pH and the intramembrane protease complex γ-secretase trigger membrane insertion and protrusion of L2 through the vesicular membrane into the cytosol for recruitment of sorting nexins and retromer [18–22]. A C-terminal charged region of L2, with inherent "cell penetration peptide" properties is also important for protrusion of L2 across the endosomal membranes [23]. In this manner L2 dictates the retrograde trafficking of L2/vDNA complexes from endosomes to the *trans*-Golgi network (TGN), an obligate step of initial infection [24,25]. Membrane-bound vDNA, still in complex with L2, exits from the vesiculating TGN at the onset of mitosis and migrates towards metaphase chromosomes where L2 directly binds, tethering the vDNA to host chromosomes to ensure efficient nuclear delivery [26,27]. As daughter cells return to G1, chromosomes decondense and PML bodies are recruited to the L2/vDNA, a process that appears to be necessary for efficient viral transcription [28,29].

HPV has evolved to infect and replicate in differentiating cutaneous or mucosal epithelium, and the viral life cycle is tightly intertwined with cellular differentiation of this tissue. Initially, HPV infects undifferentiated basal keratinocytes, resulting in low copy number maintenance of the episomal vDNA [3,30,31]. Viral replication is achieved through the coordinated expression of viral early and late genes in response to cellular differentiation, resulting in vDNA amplification, expression of L1 and L2 capsid proteins, and assembly of progeny virions [3]. Mature virions exist in an oxidized state, with intercapsomeric disulfide bonds stabilizing the particle by crosslinking of neighboring L1 pentamers [9,32,33]. Pseudovirus particles generated through the 293TT system [34–36] must be "matured" in vitro to achieve this oxidized state but virions generated through organotypic raft cultures achieve this stabilized state through a naturally occuring redox gradient in the epithelial tissue where cells of the basal and suprabasal layers contain abundant free thiols and are in a reduced state relative to the upper cornified layers of the tissue [34,37].

Homeostasis of cellular thiol and disulfide redox is largely maintained by a large intracellular pool of glutathione (GSH). Given natural redox gradient of the epithelium, and the prominent role of GSH in maintaining redox balance [38] we sought to investigate the role of cellular GSH in HPV16 infection. We find that siRNA knockdown of key enzymes in the GSH synthesis pathway impairs HPV16 pseudovirus infection. Depletion of the intracellular GSH pool caused a marked decrease in the infection of HPV16 but not adenoviral vectors. GSH was not important for HPV16 binding, endocytic uptake, cleavage of minor capsid protein L2, or trafficking of vDNA to the TGN, but was critical for efficient post-Golgi trafficking and intranuclear delivery of HPV16 L2/vDNA. Further work will be necessary to specifically define the GSH-dependent factors necessary for HPV16 infection.

## Materials & methods

### Cells and viruses

HaCaT cells, an immortalized keratinocyte line [39] and HaCaT-GFP-BAP cells that stably express cytosolic GFP-BAP (27) were cultured in high-glucose cDMEM media supplemented with 10% fetal bovine serum (FBS), and antibiotic/antimycotic (Ab/Am, Gibco #15240062). Additional 200ng/mL puromycin (EMD Millipore 508838) was used for maintaining HaCaT-GFP-BAP cells. 293TT cells were cultured in high-glucose cDMEM media supplemented with 10% bovine growth serum (BGS), Ab/Am, and 165µg/ml hygromycin B (Thermo 10687010). HPV16 PsV encapsidating the luciferase expression plasmid pGL3-basic were generated as previously described [40]. Briefly, 293TT cells were co-transfected with a pXULL-derived plasmid expressing both L1 and L2, and Transfected cells were harvested at 48hrs post-transfection by trypsinization and pelleted/resuspended in PBS + 9.5 mM $MgCl_2$ at 100µl/10cm plate, followed by the addition of Brij58 to 0.35%, ammonium sulfate (pH = 9.0) to 25mM, Benzonase nuclease (Sigma E1014) to 0.3%, and 20U/mL exonuclease V (Epicentre E3105K) with overnight incubation at 37˚C to promote maturation of capsids. After maturation, 0.17 volumes of 5M NaCl was added and lysates were 1x freeze/thawed at -80/37˚C to further break apart cellular structures. Lysates were cleared by centrifugation at 3000x g and supernatants were loaded onto discontinuous CsCl gradients made from 4ml light (1.25g/ml) CsCl underlaid with 4ml heavy (1.4g/ml) CsCl. Virions were purified by 18h ultracentrifugation at 20,000 rpm at 4˚C in Beckman SW41 Ti rotor/buckets. Viral bands were visible slightly above the gradient interface and were collected by side puncture with a 1.5" 18 gauge needle and 5ml syringe. Virions were washed 3x and concentrated in VSB (25 mM HEPES pH 7.5, 0.5M NaCl, 1 mM $MgCl_2$) using 100,000 MWCO centrifugation filter units (Sartorius VS04T42) and stored at -80˚C. SYBR green (Thermo Fisher K0252) qPCR was used to determine the packaged pGL3 copy number. The capsid/genome ratios were all within the normal range for typical HPV16 preps. To generate 5-ethynyl-2'-deoxyuridine (EdU)-labeled virus, 15µM EdU (Invitrogen C10337) was supplied in 293TT cell cultures before, during, and after the transfection of L1, L2 expressing plasmid and luciferase-expressing plasmid.

### SDS-PAGE and western blotting

For denaturing/reducing polyacrylamide gel electrophoresis (PAGE), samples were lysed in either RLB (Reporter Lysis Buffer, Promega E397A) or RIPA lysis buffer (50mM Tris-HCl PH = 8.0, 150mM NaCl, 1% Triton X-100, 0.5% Na-deoxycholate, 0.5% SDS, supplemented with 1% PMSF (Sigma 78830) and 1% protease inhibitor cocktail (Sigma P8340), combined with 20% total volume of denaturing/reducing SDS-PAGE buffer (contain 0.5M Tris, glycerol, 10%SDS, 2-mercaptoethanol and 1% bromophenol blue). Samples were then heated to 95˚C for 5 minutes prior to separation on 10% or 12.5% polyacrylamide gels and run at 110V in gel running buffer (1X Tris-Glycine-SDS) for 90–105 minutes. Samples were transferred onto nitrocellulose membranes under 300mA for 75 minutes by using 1X Western transfer buffer (0.25M Tris, 1.92M Glycine) supplemented with 10% methanol. Membranes were blocked with 5% non-fat milk in 1XTBST (200mM Tris, 1.5M NaCl, 1% Tween-20, pH = 7.5) at 4˚C overnight. For translocation experiments to detect GFP biotinylation, membranes were blocked in Odyssey Blocking Buffer TBS (LI-COR) at 4˚C overnight. For the denaturing/nonreducing PAGE, samples lysed by 1X RIPA lysis buffer supplemented with 1% PMSF, 1% proteinase inhibitor and 2mM N-ethylmaleimide (NEM, Sigma E1271) were mixed with 20% total volume of denaturing /non-reducing SDS-PAGE buffer (contain 0.5M Tris, glycerol, 10%SDS and 1% bromophenol blue). Samples were then incubated at room temperature for 10 minutes prior to PAGE.

## Western blot antibodies

Rabbit anti-GAPDH (Cell Signaling 2118) antibody was used at 1:5,000 dilution. Rabbit anti-GFP antibody (Clontech 632377) was used at 1:5,000 dilution. Mouse anti-L2 monoclonal K4-L2$_{20-38}$ (a kind gift from Martin Müller) antibody was diluted at 1:5,000. Mouse anti-GR (Santa Cruz sc-133159) antibody was used at 1:50 dilution. Rabbit anti-GCLm (Santa Cruz sc-22754) antibody was diluted 1:1,000. Mouse anti-GSS (Santa Cruz sc-365863) antibody was diluted 1:250. Mouse anti-HPV16 L1 (Camvir-1) (Abcam #ab128817) antibody was used at 1:5,000. All the primary antibodies were diluted in 5% milk in TBST except K4-L2$_{20-38}$, which was diluted in 1% milk in TBST. All IR680- and IR800-conjugated secondary antibodies (Fisher Scientific PI35518, PISA535521, PI35568, PISA535571) were diluted 1:10,000 in 5% milk in TBST. For translocation experiments, NeutrAvidin Dylight 800-conjugate (Thermo 22853) was used at 1:10,000 dilution in LiCor blocking buffer. Blots were imaged on the Licor Odyssey Infrared Imaging System.

## siRNA knockdown

HaCaT cells were seeded at 35,000 cells/well in 1ml siRNA media (high-glucose cDMEM supplemented with 10% FBS, antibiotic-free) in a 24 well plate. Prior to transfection, media was changed to 500µl per well Opti-MEM reduced serum medium with HEPES 2.4g/L sodium bicarbonate and L-glutamine (Gibco 31985070). Scramble control siRNA-A (sc-37007), GCLm (sc-40602), GR (sc-35505), and GSS (sc-41980), was diluted into Opti-MEM containing Lipofectamine RNAiMAX (Invitrogen #13778150) according to the manufacturer's instructions. 100µl per well siRNA-Lipofectamine complex (50nM siRNA final) was added dropwise to cells. At 16h post-transfection, cells were washed with PBS and media was replaced with siRNA media for 24h prior to viral infection assays.

## BSO treatment and infection assay

L-Buthionine-(S, R)-sulfoximine (BSO, Santa Cruz, CAS 83730-53-4) was prepared at 200mM in water. Cells were pretreated with 800µM BSO for 72h prior to experiments. 50,000 cells/well of water or BSO-treated HaCaTs were seeded in 24 well plate the day before infection. Cells were infected with HPV16 at $2 \times 10^8$ viral genome equivalents (pGL3 copies) per well or 10,000 luciferase expressing HAdV-5 vector particles per cell. At 24h post-infection, cells were lysed in 100µl RLB. 100µl luciferase assay reagent (Promega E1483) was added into 20µl cell lysate and luciferase activity was measured using a DTX800 Multimode plate reader (Beckman Coulter). The remainder of the cell lysate was collected for western blots and GAPDH immunostaining. GAPDH bands were quantified by densitometry using ImageJ software [41] to normalize the luciferase data.

## Glutathione measurements

GSH/GSSG-Glo assay kit (Promega, Cat.V6611) was used for glutathione measurements. After 48h 800µM H$_2$O or BSO pretreatment, 10,000 H$_2$O-treated HaCaTs cells/per well or 15,000 BSO-treated cells/per well were transferred into white 96 well plates. Then, another 24h H$_2$O or 800µM BSO treatment was followed. To measure total GSH levels, 50µl per well total glutathione lysis reagent (containing Luciferin-NT, passive lysis buffer) was added to the cells. To measure oxidized GSSG, 50ul per well oxidized lysis reagent (containing Luciferin-NT, passive lysis buffer, and 25mM NEM) was added. Plates were incubated at room temperature and shaken for 5 minutes, then 50µl per well luciferin generation reagent (containing 100mM DTT, glutathione S-transferase and GSH reaction buffer) was added. The plate was shaken

and incubated at room temperature for another 30 minutes. Finally, 100μl/well luciferin detection reagent was added and the plate was shaken and incubated at room temperature for another 15 minutes. The luminescence was measured using a DTX-800 multimode plate reader (Beckman Coulter). Standard curves were generated using serial dilutions of standard GSH (provided by the kit) ranging from 16μM to 0.25μM.

## Binding and entry assays

BSO or $H_2O$ treated cells were incubated on ice for 20 minutes before infecting with 1μg L1/ml of HPV16 in cold cDMEM media supplemented with 10% FBS. Plates were kept on ice for 1h to allow viral particles to bind to the cell surface. For the binding experiments, cells were washed with cold PBS (pH = 7.4) to completely remove the virus from the media. The control groups were washed with cold high-pH PBS (pH = 10.75) followed by regular cold PBS to remove surface bound virus. Lysates were then collected for non-reducing SDS-PAGE. For entry experiments, cells were washed with PBS, replaced with fresh media, and incubated at 37˚C in 5% $CO_2$ incubator after 1h virion pre-binding. After 2h incubation, cells were washed with high-pH PBS to remove the surface-bound virus and replaced with fresh media and incubated at 37˚C. Samples were collected at the indicated times and processed for non-reducing SDS-PAGE.

## Furin cleavage experiments

HaCaTs cells were pretreated with either 800μM BSO or $H_2O$ as described above. After 48h pretreatment, 90,000/well of treated cells were seeded on 12 well plates followed by another 24h of BSO/$H_2O$ treatment. Cells were then infected with 800ng L1/well of HPV16 virions containing the PSTCD-L2 fusion [42]. At 16h post-infection, samples were lysed with RIPA lysis buffer supplemented with 1% PMSF, 1% proteinase inhibitor and 20% denaturing reducing SDS loading buffer. Western blot was performed as previously described. The intensity of the uncleaved band and the cleaved band were quantified by densitometry using ImageJ software [41] and the fraction of cleaved L2 was calculated.

## Immunofluorescence staining

100,000 cells/well of $H_2O$ or BSO treated HaCaTs cells were plated on coverslips in 6 well plates. Cells were infected with 500ng L1/ml of virus. At 2h and 8h post-infection, cells were fixed with 2% paraformaldehyde (pH = 7.4, Fisher Scientific 30525-89-4), permeabilized by 0.25% Triton X-100 (Fisher Scientific 9002-93-1) and blocked by using blocking solution (PBS plus 4% bovine serum albumin, fraction V, Fisher Scientific 9002-46-8, supplemented with 1% goat serum). Rabbit anti-HPV16 polyclonal antibody (a kind gift from M. Ozbun) was used at 1:1,000 dilution and mouse L1-7 antibody (a kind gift from M. Sapp) was used at 1:50 dilution. Cells were incubated with primary antibody at room temperature for 1h, followed by 1h room temperature incubation in 1:1000 dilution of secondary goat anti-mouse AlexaFluor-555 antibodies (Invitrogen A21424) and goat anti-rabbit 488 antibodies (Invitrogen A11034). All antibodies were diluted in regular PBS contain 20% blocking solution. Prolong diamond anti-fade mounting medium with DAPI (Life Technologies P36971) were used for mounting coverslips. For EdU experiments, Cells were infected with 1μg L1/ml of EdU labeled HPV16 for 20h. Then, cells were washed and replaced with fresh media for another 8h incubation. Cells were then washed with high-pH PBS (pH = 10.75) followed by regular PBS to remove all the surface-bound virus. Cells were then fixed, permeabilized, and blocked as described above. Click-iT EdU AlexaFluor-488 kit (Life Technologies C10337) was used for labeling the EdU according to the manufacturer's protocol. Primary antibody and secondary antibody incubation were

then performed as described above. Rabbit anti-TGN46 (Sigma-Aldrich T7576) antibody was used at 1:200 dilution. Mouse anti-p230 (BD Transduction Laboratories, #611280) was used at 1:400 dilution. Goat anti-Rabbit AlexaFluor-555 secondary antibody (Invitrogen A21429) was used at 1:1,000 dilution.

### Confocal microscopy

Confocal microscopy was performed using a Zeiss LSM510 META system or the Zeiss LSM880 system with 405nm, 488nm, and 543nm lasers. Samples were examined using a 63x objective, and Z-stacks with a 0.35μm depth per plane were taken of each image. Representative single-plane images were processed with the Zeiss META software or Zen Blue software and further processed with ImageJ software [41].

### Colocalization analysis

Manders overlap coefficients [43] for a variety of channels within individual Z-stacks were determined using the JACoP plugin [44] on ImageJ. Thresholds were manually set below saturation. Individual Manders coefficient values and mean values from multiple Z-stacks (each containing multiple cells), across 2–5 independent experiments, were plotted with GraphPad Prism software. p-values were calculated with Prism using a two-sample unpaired *t*-test as recommended for colocalization analysis [45].

### Translocation experiments

60,000 cells/well of $H_2O$ or BSO treated HaCaT-GFP-BAP cells were seeded in 24-well plate. Cells were infected with 150ng L1/well of HPV16 L2-BirA virus [27]. At 24h post-infection, samples were processed for reducing SDS-PAGE followed by western blot to detect total and biotinylated GFP.

### PI staining and flow cytometry

$H_2O$ or BSO treated HaCaTs cells were trypsinized and pelleted. The cell pellet was washed and resuspended in cold 70% ethanol. Cells were then pelleted again and resuspended in 500ul cold PBS. Samples were incubated at 37°C for 30 minutes under 1/20 volume of RNase A at 20mg/ml in TE buffer (50mM Tris-HCl pH 8.0, 10mM EDTA), and 1/40 volume of 1.6mg/ml propidium iodide. Samples were analyzed on a BD Biosciences FACSCanto II flow cytometer using Diva 8.0 software. Counts for G1, S, or G2/M phases were plotted as percentages of cell count on MS Excel.

### Statistics

p-values were calculated with the appropriate two-sample paired or unpaired *t*-tests using MS Excel or GraphPad Prism software, as indicated. A significance threshold value of $p < 0.05$ was applied.

## Results

### Knockdown of GSH biosynthesis enzymes partially blocks HPV16 infection

Several enzymes are important in regulating GSH biosynthesis and maintaining a proper GSH (reduced)/GSSG(oxidized) ratio (Fig 1A) [46]. Glutamate cysteine ligase (GCL, EC 6.3.2.2) is a rate-limiting enzyme in GSH biosynthesis pathway, responsible for generating the dipeptide γ-glutamylcysteine (γ-GC) from intracellular cysteine and glutamate [47]. Glutathione

synthetase (GSS, EC 6.3.2.3) then adds glycine to γ-GC, forming GSH [48]. Cells normally maintain GSH at high concentrations, ranging from 1-10mM. Such a high concentration of reduced GSH protects cells from oxidative stress and reactive oxygen species (ROS), through the actions of free GSH and GSH-dependent glutathione peroxidases and glutaredoxins [49,50]. Glutathione reductase (GR, EC 1.8.1.7) maintains a high GSH/GSSG ratio by converting oxidized GSSG back to two molecules of reduced GSH [51]. This high concentration of reduced GSH maintains the cytosol in a reduced state, favoring free thiols over disulfides.

To understand whether GSH and a reducing cytosolic environment is important for HPV16 infection, we targeted the enzymes GCL, GSS and GR for siRNA knockdown and measured HPV16 infection in HaCaT cells. The heterodimeric enzyme GCL is comprised of two protein subunits; the catalytic GCLc and the modulatory GCLm [52]. Maximal catalytic activity of GCL holoenzyme is only achieved upon binding of the modulatory GCLm to the catalytic GCLc subunit. GCLm is expressed at lower levels than GCLc, and is therefore rate-limiting in the formation of active GCL holoenzyme [53]. Thus, we chose to target the GCLm subunit for siRNA knockdown experiments.

HPV16 infection dropped about 40–50% upon knockdown of either the modulatory subunit of GCL or GR (Fig 1B and 1C), suggesting that the cytosolic GSH may be important in facilitating the HPV16 infection. To our surprise, cells became slightly more permissive to HPV16 infection upon knockdown of GSS (Fig 1B and 1C). Previous research suggests that the intermediate thiol-containing metabolite γ-GC can substitute for GSH to help remove reactive oxygen species [54,55]. The slight increase of the HPV16 infection in GSS knockdown cells may be due to substitution of γ-GC for GSH.

## GSH is important for HPV16 to establish efficient infection

To further verify importance of cytosolic glutathione in HPV16 infection, HaCaT cells were treated with L-buthionine-(S,R)-sulfoximine (BSO), an irreversible and specific transition-state inactivator of GCL [56], prior to HPV16 infection. As treatment with BSO only blocks nascent GSH synthesis, time is required for a drop in cytosolic levels of GSH to be observed [57]. After 72h treatment with 800μM BSO, cytosolic GSH levels were drastically depleted compared to the control vehicle-treated cells (Table 1). BSO treated HaCaT cells displayed a slightly slower proliferation rate (Fig 2A) and cell cycle analysis by PI staining revealed a subtle expansion of S phase upon BSO treatment (Fig 2B), but neither of these differences were statistically significant ($p>0.05$). This is consistent with other studies that BSO treatment does not considerably alter cell cycle kinetics [58,59]. HPV16 infection levels dropped about 70% upon BSO treatment (Fig 2C), again suggesting that cytosolic GSH is required for efficient HPV16 infection. To rule out pleiotropic effects of BSO on luciferase expression, control or BSO treated HaCaT cells were infected with a luciferase-expressing ΔE1/E3 adenovirus (HAd5-Luc) and luciferase was measured 24h post infection. In this case, BSO actually enhanced HAd5-luc infection (Fig 2D), implying that the inhibition of HPV16 is not due to some non-specific effects of BSO on cellular viability, endocytosis, general transcription, or activity of the luciferase enzyme itself. Although BSO treatment did subtly but reproducibly affect cell proliferation and cell cycle, these changes are too minute to account for the observed decrease in HPV16 infection.

## Effects of GSH depletion on viral binding, entry, and uncoating

To determine the reason for inefficient HPV16 infection upon GSH depletion we performed binding and entry studies. We first investigated HPV16 binding to control or BSO treated HaCaT cells. Pre-chilled cells were infected with HPV16 for 1h at 4°C to prevent viral

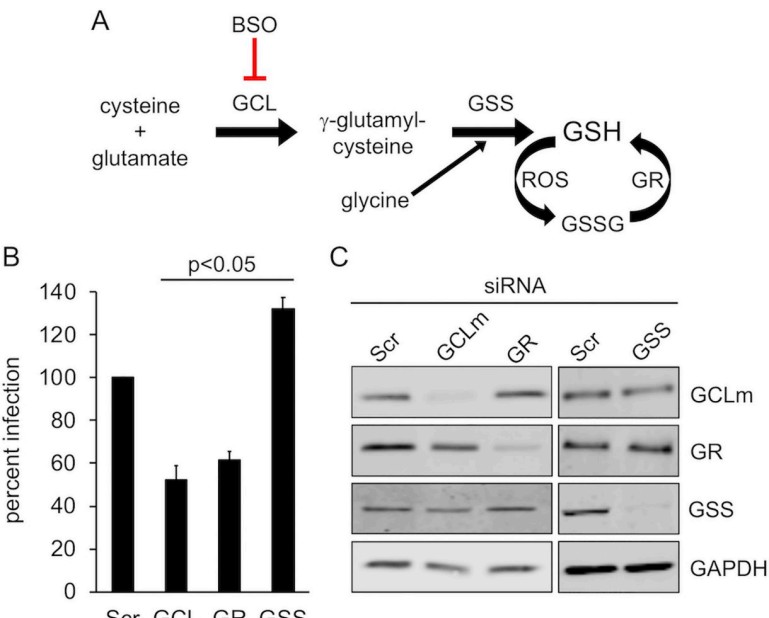

**Fig 1. siRNA knockdown of GSH biosynthetic enzymes. (A)** Schematic of GSH biosynthesis pathway. GCL catalyzes the ligation of cysteine and glutamate to generate γ-GC. Glycine is then added to γ-GC by GSS to form GSH, which can reduce ROS to form oxidized GSSG. GR maintains high levels of GSH by reducing GSSG back to GSH. BSO specifically blocks GCL, the rate-limiting enzyme in GSH biosynthesis pathway. **(B)** Infectivity of luciferase-expressing HPV16 in siRNA treated HaCaT cells. Cells were transfected with SCR, GCLm, GR or GSS siRNA for 18h prior to infection. Luciferase assays were performed 24h post infection and normalized to GAPDH expression level. The graph shows mean percent infection (±SEM, n = 3 independent experiments) with Scr infection levels set to 100%. P-values were determined with a two-sample paired t-test. **(C)** Western blot to verify specific siRNA knockdown of GCLm, GR, and GSS.

endocytosis [60] and unbound virus was extensively washed away. Some groups received an additional wash in PBS buffer at pH = 10.75 to remove any cell-bound virus as previously reported [16]. Cell lysates were then collected, and bound L1 was detected by non-reducing SDS-PAGE and western blot. Similar levels of full-length disulfide-linked L1 dimers and tri-mers were observed in control or BSO treated samples that did not receive the high pH washes (Fig 3A and 3B), indicating that GSH depletion had no effect on HPV16 binding and the high pH wash does indeed remove extracellular virus.

Next we investigated viral uptake by pre-binding HPV16 to cells followed by a 2h shift to 37˚C to allow entry. After 2h cells were either processed for non-reducing L1 western blot or cleared of extracellular virus by high pH washing and returned to 37˚C for additional times. In this manner we are able to track the population of intracellular HPV16 that has entered the

**Table 1. Measurement of GSH concentrations in cells ± BSO.**

| Cell type/condition | [GSH]* | [GSSG]* | GSH/GSSG |
|---|---|---|---|
| HaCaT + H$_2$O | 7.245 ± 1.19 | 0.094 ± 0.008 | 77.1 |
| HaCaT + BSO | 0.022 ± 0.008 | N.D. | ---- |
| HaCaT-GFP-BAP + H$_2$O | 7.548 ± 1.25 | 0.104 ± 0.01 | 72.6 |
| HaCaT-GFP-BAP + BSO | 0.018 ± 0.001 | N.D. | ---- |

*mean [μM] ±SEM, n = 2 independent experiments.

N.D. = not determined, below the limit of detection.

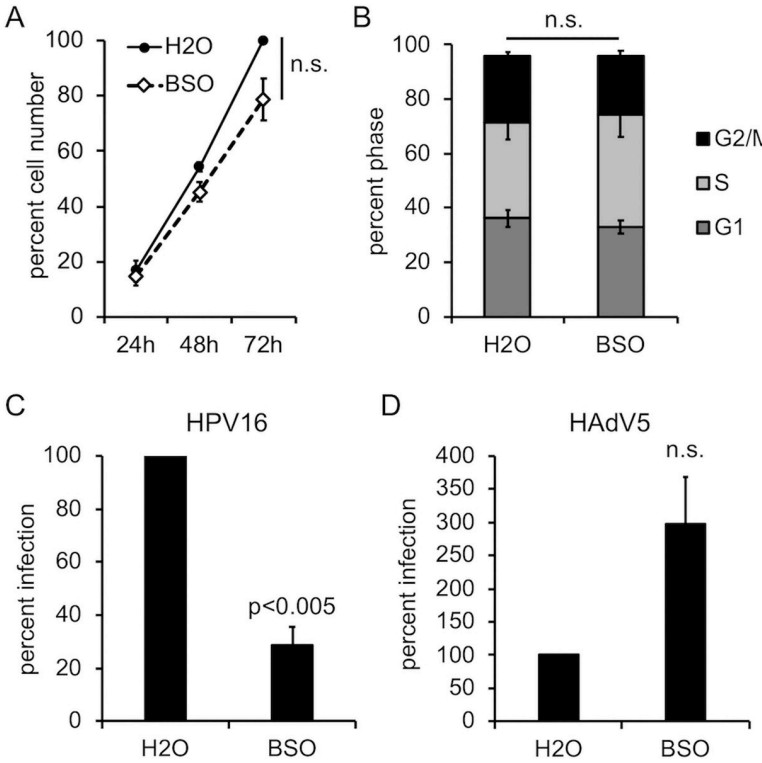

**Fig 2. GSH is important for efficient HPV16 infection. (A)** Growth curve of HaCaT cells ± BSO. The mean percent total number of cells is shown (±SEM, n = 3 independent experiments). The 72h $H_2O$ treated group was set to 100. **(B)** Cell cycle progression analysis in $H_2O$ and BSO treated HaCaT cells, as measured by PI staining and flow cytometry. The graph shows the mean percent cell counts in G1, S, or G2/M phases (±SEM, n = 3 independent experiments). **(C, D)** Infectivity of luciferase-expressing HPV16 **(C)** or luciferase-expressing HAdV5 **(D)** in HaCaT cells ± BSO. Luciferase assays were performed 24h post infection and data were normalized to GAPDH expression levels. The graph shows mean percent infection (±SEM), relative to $H_2O$ treated cells, n = 4 for **(C)**, n = 3 for panel **(D)**. P-values were determined with a two-sample paired *t*-test, with significance cut off value set to p = 0.05.

cells during the initial 2h incubation. Upon return to 37˚C this "2h wave" of incoming virus will continue trafficking through the degradative endolysosomal system. Size shifts of L1 in western blots are thereby indicative of intracellular L1 cleavage and degradation during post-entry trafficking. Analysis of entry in this manner revealed no major differences between control and BSO treated HaCaT cells, aside from a higher level of degraded L1 product with BSO treatment at the 2h timepoint (Fig 3C, compare lanes 1 and 2). Densitometry of three separate experiments revealed that both the water and BSO treated groups accumulated similar levels of high molecular weight (HMW) L1 within the initial 2h of uptake, and both groups showed similar kinetics of L1 degradation as evidenced by disappearance of the HMW L1 and concomitant appearance and disappearance of the smaller 25 kDa L1 cleavage product (Fig 3D and 3E).

Additional immunofluorescence and microscopy experiments were performed to investigate viral trafficking and disassembly within control and BSO treated cells. After HPV16 uptake, resident proteases within the acidic endosomal compartments promote the breakdown and disassembly of the L1 capsid. Although this degradative uncoating is not an obligate step of the infectious trafficking pathway, it does serve as a proxy for endolysosomal trafficking [12,61,62]. Exposure of the L1-7 epitope, which is buried within the intact capsid [63], is a useful marker for degradative uncoating and normally occurs by 6-8h post infection [64]. Control

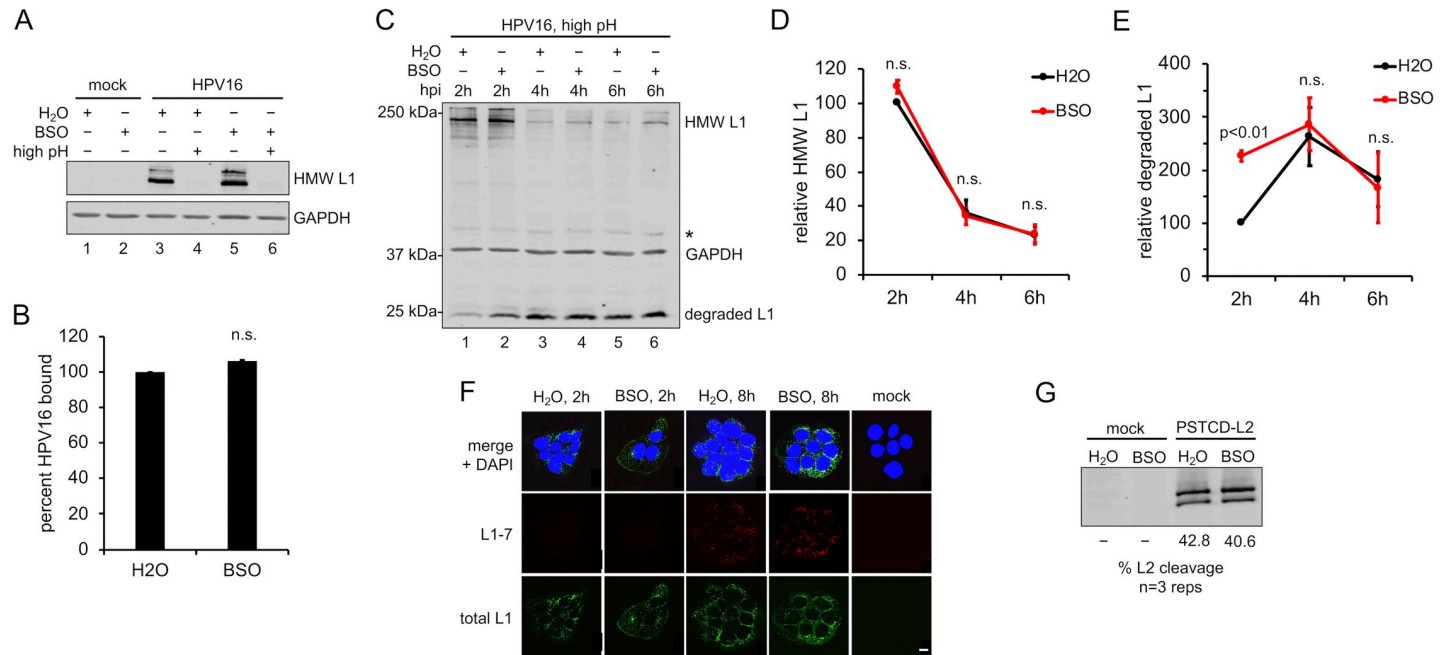

**Fig 3. GSH depletion does not perturb early events of HPV16 infection.** (A) Binding assay. HPV16 was bound to HaCaT cells ± BSO for 1h at 4°C. Cells were washed with regular PBS (pH = 7.2, lanes 1, 2, 3, and 5) or high pH PBS (pH = 10.75, lanes 4 and 6) to remove the surface-bound virus before detection of cell-bound L1 by non-reducing SDS-PAGE and western blot. (B) Densitometric quantification of band intensities from binding assays. The graph shows mean percent L1 band intensity (±SEM, n = 2 independent experiments) relative to the control group. P-values were determined with a two-sample paired *t*-test, with significance cut off value set to p = 0.05. (C) Representative HPV16 uptake assay. HaCaT cells ± BSO were prebound with virus and incubated for 2h at 37°C, at which time cell surface virus was removed by high pH PBS wash. Media ± BSO was replaced and infected cells were incubated at 37°C for additional times (4h and 6h total) to allow trafficking of intracellular virus. Cell lysates were processed for nonreducing SDS-PAGE and western blot to detect intact and degraded forms of L1. *Asterisk marks a cellular protein that cross-reacts with the L1 antibody, and serves as an internal loading control. (D, E) Densitometric quantification of intact HMW L1 (D) and degraded L1 (E) band intensities from uptake assays, as in (C). Graphs show mean relative L1 band intensities at given times (±SEM, n = 3 independent experiments) relative to the control water-treated group at the 2h time point. P-values were determined with a two-sample paired *t*-test, with significance cut off value set to p = 0.05. (F) Intracellular uncoating and degradation of HPV16 capsid. HaCaT cells ± BSO were plated on coverslips and infected with HPV16 for 2h or 8h prior to fixation and processing for IF as described in *Materials & Methods*. Total L1 was stained with rabbit anti-L1 polyclonal antibody and anti-rabbit secondary antibody (green). Degraded L1 was stained with mouse L1-7 monoclonal antibody and anti-mouse secondary antibody (red). Nuclei were stained with DAPI (blue). Representative micrographs are shown, scale bars = 10μm. (G) L2 furin cleavage assay. HaCaT cells ± BSO were infected with PSTCD-L2 virus for 16h prior to SDS-PAGE and western blotting. L2 band intensities were quantified by densitometry using ImageJ and percent L2 cleavage was calculated from three biological repeats.

and BSO treated cells were infected with HPV16 for 2h or 8h and total L1 or L1-7 levels were visualized by immunofluorescence staining and confocal microscopy. L1-7 was undetectable in either group at the early 2h time point, as expected (Fig 3F). Both the control and BSO treated groups displayed significant L1-7 signal by 8h, indicative of capsid degradation and proper endolysosomal trafficking (Fig 3F). Taken together these data suggest that although GSH depletion may result in accelerated early degradation of L1 capsid, there appears to be no major effects on HPV16 binding, entry, or endosomal trafficking and capsid degradation.

## Depletion of GSH does not affect furin cleavage of L2

During HPV16 infection, the N-terminal 12 amino acids of L2 are removed by furin cleavage [42], an obligatory step for proper retrograde trafficking and infection (reviewed in [22]). To determine if BSO affects furin processing of L2 we utilized a previously described system to easily monitor L2 cleavage called PSTCD-L2 [42]. Since a 12 amino acid difference is difficult to resolve by SDS-PAGE, the 70-residue *Propionibacterium shermanii* transcarboxylase domain (PSTCD) is fused on the N-terminus of L2 to generate a 9 kDa size shift upon furin cleavage. Control or BSO treated cells were infected with PSTCD-L2 virus and levels of L2

cleavage were measured at 16h post infection by western blot of cell lysates and staining for total L2. No significant differences were observed in the amount of cleaved L2 (Fig 3G), indicating that cytosolic GSH does not affect furin processing of L2.

## GSH is important for L2/vDNA post-Golgi trafficking and translocation

HPV16 must transport its viral genome into the cell nucleus to successfully establish infection. This process involves trafficking of L2/vDNA from endolysosomal compartments to the *trans*-Golgi where upon mitosis post-Golgi vesicle-bound L2/vDNA tethers to mitotic chromosomes for nuclear localization and ultimately penetrates the limiting membrane to gain access to transcriptional machinery [26,27,65]. To assess Golgi and nuclear localization of L2/vDNA, we infected control or BSO treated HaCaT cells with virions containing either EdU-labeled vDNA or L2-3xFlag. Cells were infected for 24h, surface virus was removed by high pH washing, and infection was continued for 8h prior to fixation, permeabilization, and staining. BSO (or water control) was included in the media for the duration of the infection. Cells were stained for either EdU (vDNA) or 3xFlag along with TGN46, a *trans*-Golgi marker. In some EdU staining experiments, p230 was used as the counterstain for the *trans*-Golgi. Nuclei were stained with DAPI. In control groups most of the EdU-labeled vDNA or L2-3xFlag was found overlapping with DAPI, within the cell nucleus (Fig 4A, 4B and 4C). In contrast, BSO treatment resulted in the retention of the vast majority of EdU labeled vDNA or L2-3xFlag within the *trans*-Golgi, as seen by overlap with either TGN46 or p230 (Fig 4A, 4B and 4C). Manders overlap coefficients [43] were measured from multiple Z-stacks across replicate experiments and differences were found to be statistically significant (Fig 4D). These data indicate that while L2/vDNA was efficiently transported to the *trans*-Golgi, GSH depletion prevented efficient exit from this compartment.

In prior work we developed a system to monitor translocation of L2 across limiting membranes [27]. The platform is based on the C-terminal fusion of the biotin ligase BirA to L2. BirA specifically biotinylates the BAP (biotin acceptor peptide) substrate [66]. Upon passage of L2-BirA across limiting membranes in reporter HaCaT-GFP-BAP cells, BirA will encounter and biotinylate the substrate GFP-BAP providing an easy readout for translocation. We first confirmed that the HaCaT-GFP-BAP cells were efficiently depleted for GSH upon treatment with BSO (Table 1) and that this led to inhibition of HPV16 infection as observed with parental HaCaT cells (Fig 5A). Translocation experiments with L2-BirA in control or BSO treated HaCaT-GFP-BAP cells revealed that GSH is required for efficient translocation across limiting membranes (Fig 5B and 5C), in agreement with the trafficking data (Fig 4). Taken together, cytosolic GSH is important for efficient post-Golgi trafficking and translocation of L2-vDNA.

## Discussion

Here we show that GSH is necessary for efficient infection by HPV16. siRNA knockdown of the GSH biosynthetic enzymes γ-GCS and GR blocked HPV16 infection by 40–50%. Conversely, knockdown of GSS, which results in the buildup of the thiol-containing metabolite γ-GC, caused a subtle increase in HPV16 infection. Likewise complete depletion of cytosolic GSH with BSO treatment blocked HPV16 infection by 70–80%, dependent on cell type. The same treatment increased transduction by HAdV5 vectors, suggesting a block specific to HPV16. GSH depletion had no effect on binding, endocytosis, furin cleavage of minor capsid protein L2, or subcellular endolysosomal trafficking of virions or retrograde trafficking of L2/vDNA to the *trans*-Golgi.

Factors that influence or control post-Golgi transport and translocation of HPV16 L2/vDNA remain largely uncharacterized. Upon entry into mitosis vesicular bound L2/vDNA is

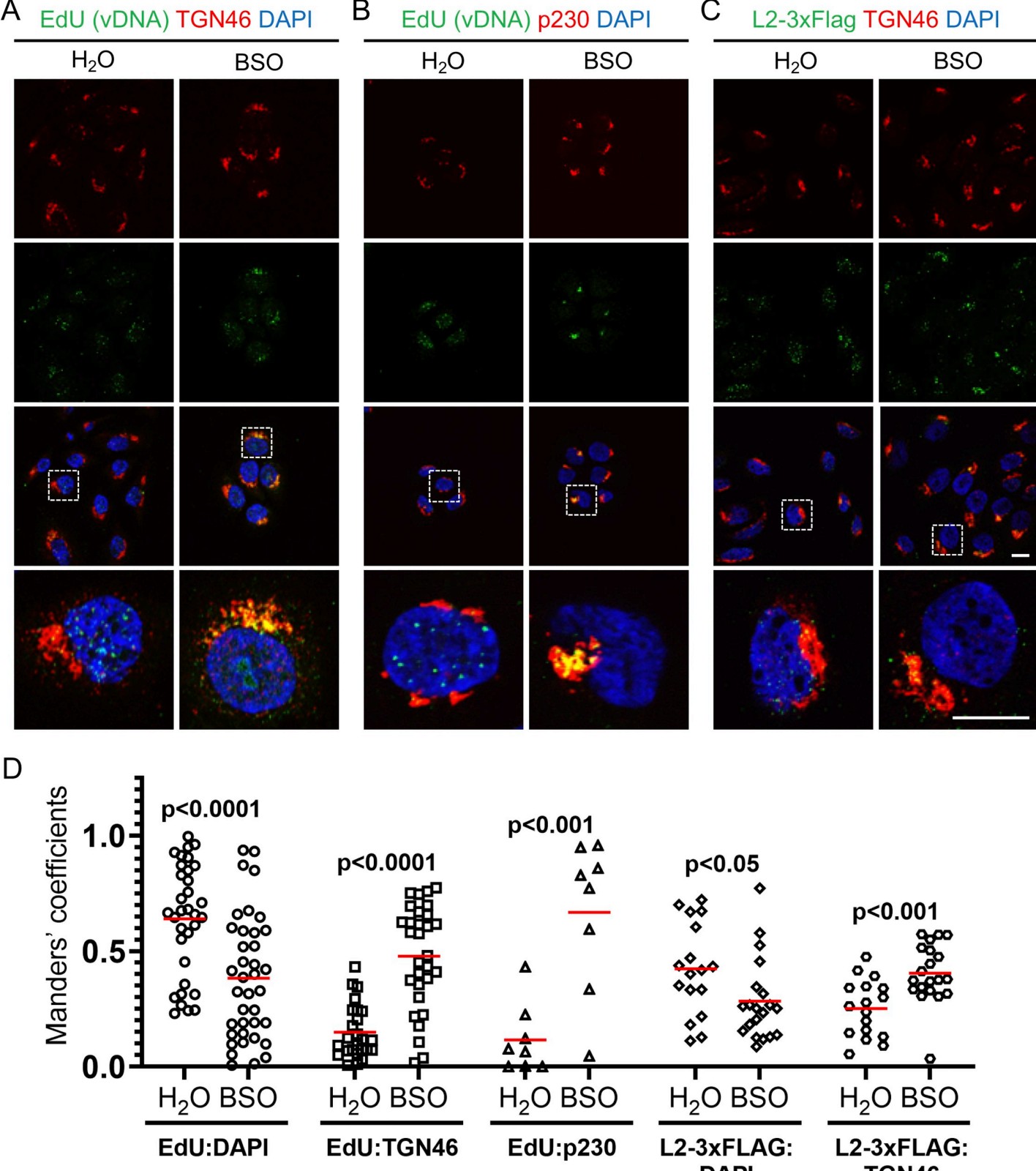

**Fig 4. Efficient L2/vDNA post-Golgi trafficking requires GSH. (A-C)** HaCaT cells ± BSO were infected with HPV16 containing either EdU-labeled vDNA or L2-3xFlag for 24h at 37˚C, followed by a high pH wash to remove surface virus, replacement of media ± BSO, and incubation at 37˚C for an additional 8h prior to fixation

and processing for IF staining as described in *Materials & Methods*. (**A**) Cells were stained with rabbit anti-TGN46 with AlexaFluor-555 conjugated anti-rabbit secondary (red), and EdU-labeled vDNA was stained with Click-iT EdU AlexaFluor-488 (green). (**B**) Cells were stained with mouse anti-p230 with AlexaFluor-555 conjugated anti-mouse secondary (red), and EdU-labeled vDNA was stained with Click-iT EdU AlexaFluor-488 (green). (**C**) Cells were stained with both mouse anti-FLAG with AlexaFluor-488 conjugated anti-mouse secondary (green), and rabbit anti-TGN46 with AlexaFluor-555 conjugated anti-rabbit secondary (red). For all panels cell nuclei were stained with DAPI (blue). Representative micrographs are shown, scale bars = 10μm. (**D**) Colocalization analysis of microscopy data using the JACoP plugin of ImageJ. Manders overlap coefficients were measured between EdU:DAPI, EdU:TGN46, EdU:p230, L2-3xFlag:DAPI, and L2-3xFlag:TGN46 for multiple Z-stacks, each containing multiple cells/field, from 2–5 independent experiments. Red bars represent the mean Manders coefficient for each data set, data points represent the Manders coefficient for each individual Z-stack. p-values were determined with a two-sample unpaired *t*-test.

believed to emanate from the scattered mitotic Golgi and traffic along the microtubule spindle towards centrioles *en route* to the mitotic chromosomes [27,65]. By metaphase, EdU labeled vDNA can be visualized bound to host chromosomes, via a central chromatin binding region [26,27]. Herein we show that cytosolic GSH is necessary for efficient egress from the Golgi, and depletion of GSH by BSO treatment inhibits HPV infection by preventing nuclear localization of L2/vDNA (Fig 6). The block could occur at the level of vesicle sorting/budding from the fragmenting Golgi, vesicular transport towards the mitotic chromosomes, or

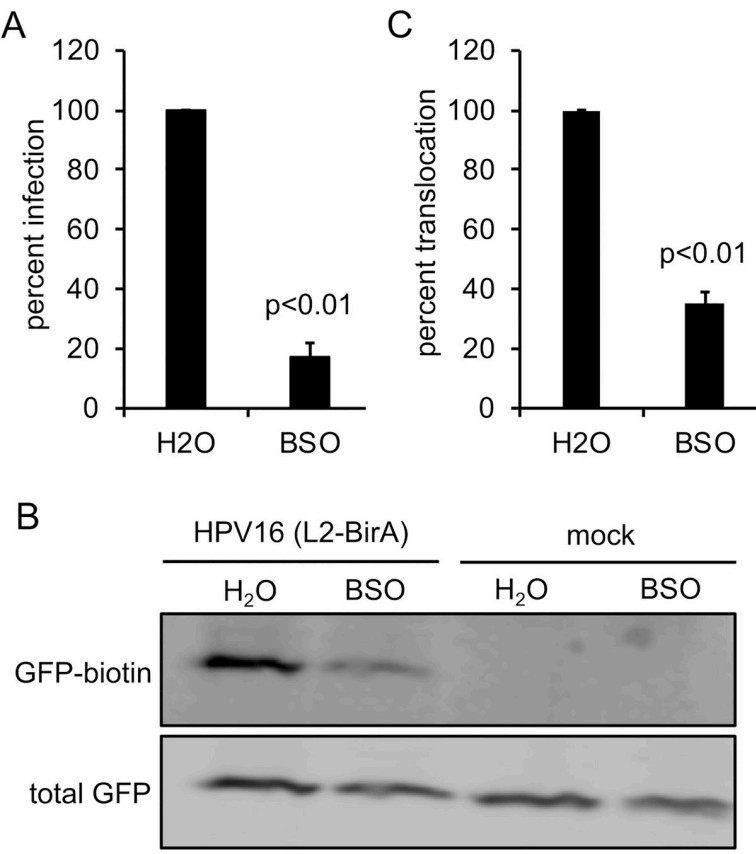

**Fig 5. GSH is required for efficient L2/vDNA translocation.** (**A**) Infectivity of HPV16 in HaCaT-GFP-BAP cells ± BSO. Luciferase assays were performed 24h post infection and data were normalized to GAPDH expression levels. The graph shows mean percent infection (±SEM, n = 3 independent experiments), relative to $H_2O$ treated cells. (**B**) Translocation assay. HaCaT-GFP-BAP cells ± BSO were infected with HPV16 containing L2-BirA. Biotinylation of GFP-BAP, a proxy for translocation across limiting membranes, was measured by SDS-PAGE and blotting with NeutrAvidin Dylight-800 conjugate. Total GFP was detected with rabbit anti-GFP antibody. (**C**) Quantification of the translocation assay blots. Band intensiies were measured by densitometry using ImageJ. The graph represents the mean percent biotin-GFP (±SEM, n = 6 independent experiments), normalized to total GFP band intensity, and relative to the $H_2O$ treated control group. For panels (**A, C**), p-values were calculated with a two-sample paired *t*-test.

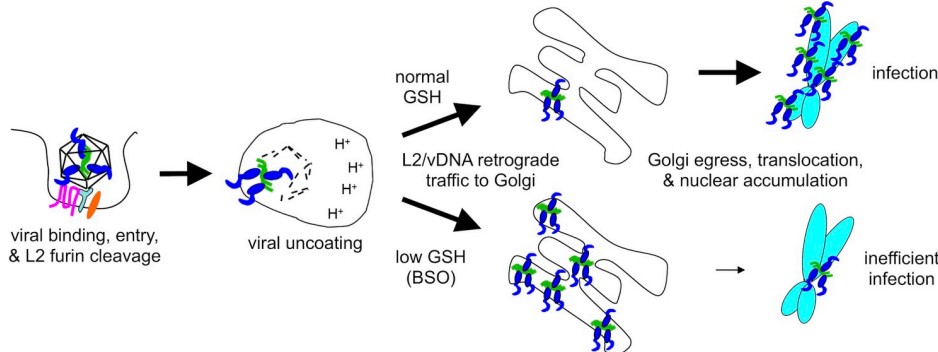

**Fig 6. Schematic summary of BSO effects.** GSH depletion by BSO treatment inhibits L2/vDNA trafficking from the Golgi to the mitotic chromosomes, resulting in accumulation of L2/vDNA within the Golgi of infected cells, low levels of nuclear L2/vDNA, and inefficient infection.

L2-chromosome interactions, all of which could conceivably affect L2/vDNA translocation and accumulation within daughter cell nuclei. Hindrance of these processes could be due to low levels of the GSH molecule itself, an altered GSH/GSSG redox couple, or perturbance of additional upstream or downstream redox/metabolite couples.

Although the precise mechanisms underlying the requirement for GSH remain to be determined, these findings are novel and significant because they represent only the second broad cellular "factor" necessary for this enigmatic process, the first being mitosis itself. While GSH did treatment did mildly perturb cellular proliferation and cell cycle, these subtle changes alone are unlikely to account for the 70–80% decreases in HPV infectivity we observed.

Cells devote energy to maintain a high intracellular concentration of GSH, which largely serves as an antioxidant to protect cells from oxidative stress and ROS. This occurs primarily in the form of GSH-dependent glutaredoxin enzymes, which use GSH to reduce protein disulfides [67], and GSH-dependent glutathione oxidase and peroxidase enzymes that catalyze the reduction of $O_2$ and $H_2O_2$ by GSH [68,69]. Free glutathione can also directly reduce oxidized disulfides [70]. Glutathione reductase is an NADPH-dependent enzyme that reduces oxidized GSSG into free GSH, maintaining a high cytosolic GSH/GSSG ratio. This high GSH/GSSG ratio ensures a reducing cytosolic redox potential, and most cytosolic sulfhydryl groups are present as free thiols rather than oxidized disulfides. These free protein thiols are therefore maintained in the reduced state and protected from harmful oxidants by excess GSH.

Proteins important for vesicular trafficking and vesicle fusion including N-ethylmaleimide (NEM) sensitive factor (NSF) and soluble NSF-attachment proteins (SNAPs) are known to be inactivated by oxidation of key cysteine residues [71,72]. The Ras, Rho/Rac, and Rab families of GTPases, key modulators of cellular signaling, cytoskeletal dynamics, organelle membrane remodeling, and vesicular transport, contain various C-terminal cysteine motifs that must be isoprenylated for proper membrane localization and function [73,74]. ADP-ribosylation factor 1 (Arf1), an important GTPase that modulates Golgi physiology and vesicular trafficking also contains a critical NEM-sensitive cysteine residue [75]. Moreover, protein S-glutathionylation and S-nitrosylation can regulate many aspects of cellular physiology, including vesicular trafficking [76,77]. Thus, it is conceivable that some critical aspects of particular vesicular trafficking pathways may require reduced cysteine residues for efficient function, and disturbing the natural GSH/GSSG couple may disrupt this trafficking. Given the complexity of vesicular trafficking and GSH physiology, elucidating the exact mechanisms through which GSH depletion affects post-Golgi trafficking of HPV16 may prove difficult.

It is interesting that the observed defect in HPV16 infection upon GSH depletion matches the phenotype of the recently described "Golgi retention" L2 mutants IVAL286AAAA, RR302/305AA, and RTR313AAA [26,78]. These mutations within the chromatin binding region of L2 prevent efficient tethering of L2 to mitotic chromosomes resulting in accumulation of vesicle-bound L2/vDNA at the Golgi compartment rather than localizing to the nucleus [26]. It may therefore be worthwhile to examine how free GSH may affect chromatin binding and localization of L2.

## Acknowledgments

We are grateful to Dr. Martin Müller for the K4-L2$_{20-38}$ monoclonal antibody, Dr. Martin Sapp for the L1-7 monoclonal antibody, Dr. Michelle Ozbun for the anti-HPV16 polyclonal antibody, Dr. Michael Barry for the HAdV5 vector, Dr. Chris Buck for the 293TT and Dr. Anne Cress for the HaCaT cells. We thank Patty Jansma of the UA ORD Imaging Core-Marley, and Paula Campbell and John Fitch of the UACC/ARL Cytometry Core Facility.

## Author Contributions

**Conceptualization:** Shuaizhi Li, Samuel K. Campos.

**Formal analysis:** Shuaizhi Li, Samuel K. Campos.

**Funding acquisition:** Samuel K. Campos.

**Investigation:** Shuaizhi Li, Matthew P. Bronnimann, Spencer J. Williams.

**Supervision:** Samuel K. Campos.

**Writing – original draft:** Shuaizhi Li, Samuel K. Campos.

**Writing – review & editing:** Shuaizhi Li, Matthew P. Bronnimann, Spencer J. Williams, Samuel K. Campos.

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
