## [Decision Letter · Decision Letter 0]

22 Oct 2019

PONE-D-19-26859

Glutathione Contributes to Efficient Post-Golgi Trafficking of Incoming HPV16 Genome

PLOS ONE

Dear Dr. Campos,

Thank you for submitting your manuscript to PLOS ONE. After careful consideration, we feel that it has merit but does not fully meet PLOS ONE’s publication criteria as it currently stands. Therefore, we invite you to submit a revised version of the manuscript that addresses the points raised during the review process.

We would appreciate receiving your revised manuscript by Dec 06 2019 11:59PM. To enhance the reproducibility of your results, we recommend that if applicable you deposit your laboratory protocols in protocols.io, where a protocol can be assigned its own identifier (DOI) such that it can be cited independently in the future. For instructions see: http://journals.plos.org/plosone/s/submission-guidelines#loc-laboratory-protocols

We look forward to receiving your revised manuscript.

Kind regards,

Craig Meyers, Ph.D.

Academic Editor

PLOS ONE

Journal Requirements:

2.  Our internal editors have looked over your manuscript and determined that it is within the scope of our Microbes & Host Cell Membrane Interactions Call for Papers. This collection of papers is headed by a team of Guest Editors for PLOS ONE: Dr Nihal Altan-Bonnet, Dr Stacey Gilk, Dr Richard Hayward, and Dr Luis Schang. The Collection will encompass a diverse range of research articles which contribute to our understanding of the mechanisms through which viruses, bacteriophage, bacteria, fungi, parasites, and microbial toxins interact with host cell and host-derived membranes.  Additional information can be found on our announcement page: https://collections.plos.org/s/microbes.

If you would like your manuscript to be considered for this collection, please let us know in your cover letter and we will ensure that your paper is treated as if you were responding to this call. If you would prefer to remove your manuscript from collection consideration, please specify this in the cover letter.

488 This work was supported by grant 1R01AI108751-01 from the National Institute for

489 Allergy and Infectious Diseases. We are grateful to Dr. Martin Müller for the K4-L220-38

490 monoclonal antibody, Dr. Martin Sapp for the L1-7 monoclonal antibody, Dr. Michelle

491 Ozbun for the anti-HPV16 polyclonal antibody, Dr. Michael Barry for the HAdV5 vector,

492 Dr. Chris Buck for the 293TT and Dr. Anne Cress for the HaCaT cells. We thank Patty

493 Jansma of the UA ORD Imaging Core-Marley, and Paula Campbell and John Fitch of

494 the UACC/ARL Cytometry Core Facility, which is funded by a UA Cancer Center

495 Support Grant (CCSG - CA 023074).

Please remove any funding-related text from the manuscript and let us know how you would like to update your Funding Statement. Currently, your Funding Statement reads as follows:  "SKC is supported by grant 1R01AI108751-01 from the National Institute for Allergy and Infectious Diseases, https://www.niaid.nih.gov/.

Reviewers' comments:

Reviewer's Responses to Questions

**Comments to the Author**

1. Is the manuscript technically sound, and do the data support the conclusions?

Reviewer #1: Yes

Reviewer #2: Yes

Reviewer #3: Yes

2. Has the statistical analysis been performed appropriately and rigorously? 

Reviewer #1: Yes

Reviewer #2: Yes

Reviewer #3: Yes

3. Have the authors made all data underlying the findings in their manuscript fully available?

Reviewer #1: Yes

Reviewer #2: Yes

Reviewer #3: Yes

4. Is the manuscript presented in an intelligible fashion and written in standard English?

Reviewer #1: Yes

Reviewer #2: Yes

Reviewer #3: Yes

5. Review Comments to the Author

Reviewer #1: Cellular entry of papillomaviruses is a complex process that is only partly understood. A critical stage of entry involves the penetration of a portion of L2 across the endosomal membrane to facilitate trafficking of vesicles containing the viral genome to the nucleus. In this paper, Li et al show that inhibition of the cellular glutathione system through drug treatment of knockdown of GSH-generating enzymes results in less efficient penetration of L2 into the cytoplasm, trafficking of the virus to the nucleus, and consequently virus infection. This is a nice paper, well executed and well presented, and the data are convincing. A few points/questions should be addressed:

1. The authors make the point (lines 326-330) that BSO treatment does not affect the cell cycle. They say that there was “a subtle but statistically insignificant expansion of S phase upon BSO treatment.” If the expansion is not statistically significant, then they cannot say that there is an expansion. It also does not help their case to claim that there is one. Better to say something like “no statistically significant changes in cell cycle profiles were observed”, and leave it at that.

2. In figure 3C, there seems to be an increase in the levels of the 25kD degraded L1 band in BSO vs control. Was this consistently observed? The authors should quantify their western results across several blots.

3. The graph in Fig 4D is kind of hard to read with the little symbols. It would be better to indicate the things being quantified with labels under the X axis.

4. In the discussion, the authors discuss ways that GSH might affect vesicular trafficking. This may important, but their own data in Fig 5 show that penetration of L2 C terminus into the cytoplasm is the key step. Failure to translocate L2 would be sufficient to explain the failure in trafficking of HPV-containing vesicles. In that light, the authors should discuss more about what is known about that translocation step. How might cellular redox affect it? What is the relationship between redox in the cytoplasm and redox in the endosomal lumen, where L2 would be present prior to translocation? Does L2 have redox-sensitive cysteines?

5. Would knockdown of the GSH production machinery affect the levels of NADPH? Is it known what effect high NADPH might have on the cell?

Reviewer #2: Overall, this manuscript provides relevant and interesting insight into what has been an experimentally challenging question – that of understanding, at a molecular level, how HPV virions enter the cell and begin their life cycle. While the insight now provided regarding how the virions exit the trans-Golgi is somewhat general, it does provide a useful framework and sets the stage for a more detailed molecular explanation in the future.

Specific comments are noted below.

• A figure showing virion entry and trafficking, highlighting the areas where glutathione/redox is likely to be important, would help the reader to navigate the manuscript.

• Have the authors definitely established that it is the specific concentrations of the actual GSH molecule that are required, or could the requirement be something more general, such as redox potential? It may be possible to provide experimental evidence one way or the other; alternatively, perhaps the language could be broadened somewhat. (For example, see page 20, lines 443-445).

Reviewer #3: This manuscript from the Campos group highlights a key role for aspects of the cellular redox system in HPV16 infection. Using compounds and siRNA depletion of key cellular factors coupled to established (and well controlled) assays they demonstrate that the role of these factors appears to occur in post-Golgi trafficking. The study is of interest to those working out the convoluted infection process of HPV. The data are tight and the manuscript well written. I do not see a need to ask the authors to perform more experiments when the present study is sufficient.

6. PLOS authors have the option to publish the peer review history of their article (what does this mean?). If published, this will include your full peer review and any attached files.

Reviewer #1: No

Reviewer #2: No

Reviewer #3: No

---

## [Author Response · Author response to Decision Letter 0]

31 Oct 2019

Response to Reviewers (responses in red) 

Reviewer #1: Cellular entry of papillomaviruses is a complex process that is only partly understood. A critical stage of entry involves the penetration of a portion of L2 across the endosomal membrane to facilitate trafficking of vesicles containing the viral genome to the nucleus. In this paper, Li et al show that inhibition of the cellular glutathione system through drug treatment of knockdown of GSH-generating enzymes results in less efficient penetration of L2 into the cytoplasm, trafficking of the virus to the nucleus, and consequently virus infection. This is a nice paper, well executed and well presented, and the data are convincing. A few points/questions should be addressed:

1. The authors make the point (lines 326-330) that BSO treatment does not affect the cell cycle. They say that there was “a subtle but statistically insignificant expansion of S phase upon BSO treatment.” If the expansion is not statistically significant, then they cannot say that there is an expansion. It also does not help their case to claim that there is one. Better to say something like “no statistically significant changes in cell cycle profiles were observed”, and leave it at that.

response: Thank you for raising this concern but we feel that these subtle effects of BSO may indeed be biologically relevant, even if they are not statistically “significant” based on the commonly used p-value threshold of 0.05. We feel it’s better to be upfront in reporting these differences and say they were not significant rather than to ignore or dismiss them, especially because these subtle trends were seen across multiple independent experiments. The text has been slightly modified to better reflect this perspective.

2. In figure 3C, there seems to be an increase in the levels of the 25kD degraded L1 band in BSO vs control. Was this consistently observed? The authors should quantify their western results across several blots.

response: Thank you for pointing this out. Further analysis of multiple replicates do indeed reveal a significant difference in degraded L1 at this 2h time point. We have performed densitometry of these reps and include that analysis as new panels in figure 3. Text has been modified accordingly.

3. The graph in Fig 4D is kind of hard to read with the little symbols. It would be better to indicate the things being quantified with labels under the X axis.

response: We have modified figure 4, and directly labeled the X-axis as suggested.

4. In the discussion, the authors discuss ways that GSH might affect vesicular trafficking. This may important, but their own data in Fig 5 show that penetration of L2 C terminus into the cytoplasm is the key step. Failure to translocate L2 would be sufficient to explain the failure in trafficking of HPV-containing vesicles. In that light, the authors should discuss more about what is known about that translocation step. How might cellular redox affect it? What is the relationship between redox in the cytoplasm and redox in the endosomal lumen, where L2 would be present prior to translocation? Does L2 have redox-sensitive cysteines?

response: Our data indicated failure of L2/vDNA to traffic post-Golgi. This could be due to failure of L2/vDNA to bind mitotic chromosomes or failure of L2/vDNA to egress/traffic from the vesiculating Golgi. We added a bit to what is known about the post-Golgi trafficking and translocation step in the discussion, lines 464-475. 

5. Would knockdown of the GSH production machinery affect the levels of NADPH? Is it known what effect high NADPH might have on the cell?

response: We did not directly look at NAD+/NADH (or NADP/NADPH) levels, but GSH depletion certainly affects these redox couples which may indeed affect many aspects of cellular physiology including DNA damage (via NAD+ dependent PARP activity), survival, and ROS responses. All these probable effects add to the challenges of understanding the molecular basis for the observed phenotype. We chose not to comment or speculate too much in this regard.

Reviewer #2: Overall, this manuscript provides relevant and interesting insight into what has been an experimentally challenging question – that of understanding, at a molecular level, how HPV virions enter the cell and begin their life cycle. While the insight now provided regarding how the virions exit the trans-Golgi is somewhat general, it does provide a useful framework and sets the stage for a more detailed molecular explanation in the future.

Specific comments are noted below.

• A figure showing virion entry and trafficking, highlighting the areas where glutathione/redox is likely to be important, would help the reader to navigate the manuscript.

response: We have added a final summary figure 6 based on the recommendation.

• Have the authors definitely established that it is the specific concentrations of the actual GSH molecule that are required, or could the requirement be something more general, such as redox potential? It may be possible to provide experimental evidence one way or the other; alternatively, perhaps the language could be broadened somewhat. (For example, see page 20, lines 443-445).

response: We currently cannot say whether it’s the GSH molecule itself, the GSH/GSSG redox potential, or downstream effects on other redox couples. We have broadened the language in the discussion to reflect this, lines 471-477.

Reviewer #3: This manuscript from the Campos group highlights a key role for aspects of the cellular redox system in HPV16 infection. Using compounds and siRNA depletion of key cellular factors coupled to established (and well controlled) assays they demonstrate that the role of these factors appears to occur in post-Golgi trafficking. The study is of interest to those working out the convoluted infection process of HPV. The data are tight and the manuscript well written. I do not see a need to ask the authors to perform more experiments when the present study is sufficient.

---

## [Editor Report · Decision Letter 1]

7 Nov 2019

Glutathione Contributes to Efficient Post-Golgi Trafficking of Incoming HPV16 Genome

PONE-D-19-26859R1

Dear Dr. Campos,

We are pleased to inform you that your manuscript has been judged scientifically suitable for publication and will be formally accepted for publication once it complies with all outstanding technical requirements.

With kind regards,

Craig Meyers, Ph.D.

Academic Editor

PLOS ONE
---

## [Editor Report · Acceptance letter]

12 Nov 2019

PONE-D-19-26859R1 

Glutathione Contributes to Efficient Post-Golgi Trafficking of Incoming HPV16 Genome 

Dear Dr. Campos:

I am pleased to inform you that your manuscript has been deemed suitable for publication in PLOS ONE. Congratulations! Your manuscript is now with our production department. 

With kind regards,

on behalf of

Prof. Craig Meyers 

Academic Editor

PLOS ONE